# The Use of Biologics for Targeting GPCRs in Metastatic Cancers

**DOI:** 10.3390/biotech14010007

**Published:** 2025-01-30

**Authors:** Cian McBrien, David J. O’Connell

**Affiliations:** School of Biomolecular & Biomedical Science, University College Dublin, D04 V1W8 Dublin, Ireland; cian.mcbrien@ucd.ie

**Keywords:** cancer, metastasis, GPCR, monoclonal antibodies, peptides, nanobodies

## Abstract

A comprehensive review of studies describing the role of G-protein coupled receptor (GPCR) behaviour contributing to metastasis in cancer, and the developments of biotherapeutic drugs towards targeting them, provides a valuable resource toward improving our understanding of the opportunities to effectively target this malignant tumour cell adaptation. Focusing on the five most common metastatic cancers of lung, breast, colorectal, melanoma, and prostate cancer, we highlight well-studied and characterised GPCRs and some less studied receptors that are also implicated in the development of metastatic cancers. Of the approximately 390 GPCRs relevant to therapeutic targeting, as many as 125 of these have been identified to play a role in promoting metastatic disease in these cancer types. GPCR signalling through the well-characterised pathways of chemokine receptors, to emerging data on signalling by orphan receptors, is integral to many aspects of the metastatic phenotype. Despite having detailed information on many receptors and their ligands, there are only thirteen approved therapeutics specifically for metastatic cancer, of which three are small molecules with the remainder including synthetic and non-synthetic peptides or monoclonal antibodies. This review will cover the existing and potential use of monoclonal antibodies, proteins and peptides, and nanobodies in targeting GPCRs for metastatic cancer therapy.

## 1. Introduction

G-protein coupled receptors (GPCRs) are the largest class of human membrane proteins with 810 receptors identified [1,2,3,4]. Of these, 455 have olfactory functions, and the remaining 355 receptors mediate the signalling of a wide variety of ligands including odours, hormones, neurotransmitters, and chemokines, that range from photons to amines, carbohydrates, lipids, and peptides to globular proteins [3,4,5]. GPCRs were originally separated into six classes, A-F based on sequence homology, of which D and E classes are not expressed in vertebrates. Those found in vertebrates were then classified into the GRAFS system which groups GPCRs together based on structural features, functionality and ligand specificity. Rhodopsin family receptors (Class A), Secretin Family receptors (Class B), Glutamate family receptors (Class C), Adhesion family receptors and Frizzled family receptors (Class F) [6]. As many as 390 GPCRs are of therapeutic interest, with the majority remaining unexplored therapeutically. GPCRs mediate a wide variety of cellular responses to stimuli and, thus, are widely involved in regulating physiological processes such as cell growth, differentiation, immune regulation, sensory, and neurological processes [7]. Aberrant signalling by these receptors can lead to and drive tumorigenic behaviour in cancer cells.

GPCRs are composed of a single polypeptide chain, with seven membrane-spanning alpha helices that combine to form a barrel-like structure that transduces extracellular stimuli across the cell membrane. The extracellular section contains the N-terminus and three extracellular loops (ECL 1–3), and is associated with ligand recognition and binding, while the intracellular section contains the C-terminus, the three intracellular loops (ICL 1–3), and mediates the signalling cascades of G proteins, kinases, and arrestins [8]. The binding of a ligand to the extracellular domain causes a conformational change, the most common being the outwards movement of transmembrane helix 6 (TM6), along with the relative shifting of the other helixes. This, in turn, exposes an intracellular pocket which allows the forming of a complex with G proteins, G-protein coupled receptor kinases (GRKs), and arrestins [9]. Each ligand can have different effects on a receptor’s signalling. Full agonists elicit a maximal signal, while partial or inverse agonists induce reduced or minimal signalling, respectively. A key aspect of GPCR signalling is biased signalling, where different ligands can stabilise distinct receptor conformations, which results in the preferential activation of specific pathways. Cryogenic electron microscopy (Cryo-EM) has been used to determine the structural basis of biased signalling and has shed further light on structural elements of ligand–receptor interactions and how these shape intracellular signalling [10].

Classical GPCR signal transduction results in the activation of the heterotrimeric G proteins which are composed of α, β, and γ subunits. The α subunit is composed of 4 families, G_s_, G_i/o_, G_q/11_, and G_12/13_. When bound to GDP, the α subunit also forms an inactive complex with the Gβγ dimer. Following receptor activation, it rapidly dissociates from GDP and binds GTP. This results in a conformational change in Gα resulting in the release of the Gβγ dimer [8,11,12]. These two subunit complexes have been shown to regulate the activity of various downstream effector proteins. The Gα subunit regulates downstream effector proteins such as DAG/IP3 and RhoGEF, while the Gβγ subunit interacts with phospholipases, ion channels, and GRKs [8,11,12]. Receptor signalling is then terminated by GRKs as they phosphorylate the carboxy terminal tail of the receptor. This recruits β-arrestin, which recognises phosphorylated receptors to which it binds and prevents further activity of the associated G proteins. It does so by occupying the same binding space as G proteins, which rapidly dissociate from the receptor in the presence of GTP, thereby regulating receptor activity [10]. It also allows for facilitation with clathrin, resulting in the endosomal degradation of the receptor [8,13]. This is the classical form of signalling, although, it is now understood that many receptors continue to signal throughout the endosomal pathway and β-arrestin can also activate alternative MAPK pathways, serine/threonine kinases, as well as c-Jun N terminal kinases [13,14]. Receptors have also been shown to have internal signalling from endosomal compartments in the Golgi apparatus [15]. Due to their extensive signalling mechanisms and their wide involvement in regulating physiological processes, mutations, or changes in the expression of these receptors can disrupt this signalling network, leading to continuous downstream signalling of pathways such as MAPK/ERK and PI3K/AKT that are involved in cell proliferation and growth. Dysregulation of these pathways can lead to the extended survival and growth of cells resulting in cancerous phenotypes [1,16,17].

Key pathways through which GPCRs can promote cancer development are the chemokine, protease-activated receptor (PAR), Hippo, and the WNT signalling pathways and many others, that mediate key tumorigenic characteristics such as cell proliferation, differentiation, immune system regulation, and migration [17]. One particular process these pathways can activate is epithelial–mesenchymal transition (EMT). This is a process by which cells transition from an epithelial phenotype to a mesenchymal phenotype as the expression of epithelial genes such as E-cadherin is decreased and mesenchymal genes, such as vimentin are increased. This causes a loss of cell–cell adhesion and an increase in stem cell-like features making these cells more invasive, thereby allowing increased metastasis [18]. These pathways can also affect the tumour microenvironment (TME), specifically through the recruitment and interaction with cancer associated fibroblasts (CAFs) which have been shown to promote cell proliferation, angiogenesis, and metastasis of cancer cells [19]. Here, we will describe the role of some key GPCRs in various metastatic cancers as well as the therapeutic application of monoclonal antibodies (mAbs), peptides, and nanobodies (the VHH variable domain of heavy chain antibodies derived from camelids).

## 2. GPCRs in Metastatic Cancers

Metastasis is perhaps the most malignant characteristic of cancers and causes 90% of all cancer related deaths [20]. Metastasis is the process by which a primary tumour mass disseminates from its original site to a new tissue niche via blood vessels or the lymphatic system, and once this occurs they become highly resistant to therapy [21]. GPCRs have been shown to have a significant role in the ability of tumour cells to metastasize with a third of the druggable GPCR family shown to promote metastatic cancers (Figure 1 and Figure 2, Table 1).

### 2.1. Metastatic Breast Cancer

Metastatic breast cancer is one of the leading causes of cancer-related deaths in women worldwide, and the role of GPCRs in this disease is a rapidly growing area of research, due to their role in key events during metastasis [214]. The chemokine receptor family for example, one of the largest class A receptor subtypes, has been extensively studied. Chemokine receptors engage in a large array of cellular functions, but their most prominent function is in cell movement. They regulate movement in different ways, including chemotaxis, haptotaxis, and transcellular migration [215,216]. Among 92 two GPCRs that have been implicated in the metastasis of breast cancer cells, CXCR4 is the possibly the best characterised, with a significant majority of published research describing the role of this receptor in metastasis, while there are many more that have had very few studies into their involvement in metastatic cancer such as NPY1R/5R or RXFP1. CXCR4 in breast cancer plays a critical role in cancer progression by promoting growth as well as allowing for metastasis to distant tissues that express its ligand CXCL12 including lung and bone niches [217]. Knocking out CXCR4 in mice delayed and regressed the growth of primary tumours, as well as preventing metastasis, showing its key role in the growth of primary tumours as well as metastasis of breast cancers [218]. HER2 activity has been shown to enhance the expression of CXCR4 as well as prevent its degradation, facilitating metastasis to the lungs [219]. More recently it has been shown that inhibiting CXCR4 with plerixafor^®^, a small molecule antagonist to the receptor, reduces fibrosis in breast cancers that have metastasized to the lungs and liver, increases T-lymphocyte infiltration, and more than doubles the sensitivity of breast cancer cells to immunotherapy [220]. Another well-studied chemokine receptor involved in breast cancer metastasis is CCR7. CCR7 has been shown to form heterodimers with CXCR4 in breast cancer cells resulting in a metastatic phenotype as well as allowing for increased survival in the absence of an extracellular matrix (ECM) attachment [221]. Silencing CCR7 in metastatic breast cancer cell lines has been shown to reduce motility, migration, and invasion both in vitro and in vivo [31,222]. PAR1 is a protease-activated GPCR whose interaction with the extracellular protease thrombin has been shown to activate breast carcinoma cells and initiate their invasion [223]. The use of two PAR1 antagonists, MMP-1 inhibitor and P1pal7, caused significant apoptosis and reduced metastasis to the lungs by 85% in xenografted mice [224]. The expression of PAR1 on breast cancer cells causes a loss of epithelial markers such as E-cadherin and gain of mesenchymal markers including vimentin, shifting them to an invasive phenotype and allowing a HMG2A mediated invasion of breast cancer [225]. PAR1 expression has been shown to be induced by the Twist transcriptional factor, which also downregulates E-cadherin expression, promoting tumour progression and metastasis. PAR1 activation leads to the downregulation of the Hippo pathway, thereby inducing an epithelial–mesenchymal transition in breast cancer [101]. ADGRF5 (GPR116) is a member of the adhesion GPCR family, the second largest of the GPCR families. They have long N-terminal adhesion regions and are involved in cell adhesion, motility, and immune response [226]. The GPCR ADGRF5 has been shown to be a regulator of breast cancer metastasis, with knock out of ADGRF5 in triple negative breast cancer cells, reducing metastasis in mouse models. ADGRF5 signalling modulates the formation of actin stress fibres and lamellipodia via Rho GTPase signalling [227]. More recent studies into the role of ADGRF5 in breast cancer metastasis showed that the loss of ADGRF5 in breast cancer cells reduced cell motility, extracellular matrix remodelling, and tumour growth. It was also shown that the loss of ADGRF5 increased the expression of MMP-8, a metalloprotease that leads to the secretion of CXCL8, resulting in increased infiltration of tumour associated neutrophils (TANs) [35]. GPER (GPR30) is a GPCR that mediates oestrogen signalling and has been shown to be significantly associated with other pro-migratory genes and metastatic pathways in ER negative breast cancer patients; high expression of GPER is also associated with lower disease-free interval in these patients [228]. GPER has also been shown to mediate oestrogen signalling in cancer associated fibroblasts contributing to migration, spreading, and the triggering of more aggressive malignant features [125]. Conversely, it has also been shown that the activation of this receptor in triple negative breast cancer using its endogenous ligand G1 reduces the angiogenesis and migration of these cells, as well as xenograft tumours [229], further showing the need to understand the temporal activation of receptors more clearly to target them therapeutically.

### 2.2. Metastatic Colorectal Cancer

Colorectal cancer (CRC) is responsible for a third of all cancer deaths in the United States. Twenty percent of all patients are diagnosed with metastatic colorectal cancer, with a 5-year survival rate of less than 5%, highlighting the lack of effective treatments in this area [230]. Colorectal cancers metastasize to many organs but predominantly the liver, lungs, and the peritoneum, with a number of GPCRs involved in driving this behaviour. CXCR4 has a well-described role in CRC, with high expression levels in patients associated with poor overall survival and progression-free survival [231]. The activation of the CXCR4/CXCL12 axis was shown to upregulate a series of miRNAs that interact with tumour associated macrophages at the invasive fronts of tumours, resulting in M2 polarisation of these macrophages. These cells then increase the metastatic capacity of CRC cells via secretion of VEGF and enhancing EMT [232,233]. The overexpression of CXCR4 has been shown to induce the formation of pseudopodia. The reorganisation of the cytoskeleton in CRC cells and activation via its ligand causes the secretion of a metalloproteinase MMP-9, increasing cell migration and metastasis [217,234]. CXCR4 interacts with CD133, a marker of CRC stem cells, in CRC metastasis. CXCR4+CD133+ cells were found in higher amounts in metastatic liver cancer, and were shown to be involved in carcinogenesis [235]. CXCR7 is active in CRC, sharing the same ligand and heterodimerizing with CXCR4. CXCR7 has been shown to be overexpressed in CRC. The gene silencing of CXCR7 inhibited growth, invasion, and induced apoptosis in CRC cells. This was due to the downregulation of PCNA, a nuclear protein and marker of cell proliferation and MMP-2, suggesting the involvement of ERK1/2 and β-arrestin signalling pathways [236]. CXCR7 activation by CXCL12 was shown to bias its signalling to β-arrestin, which promoted EMT and metastasis through induction of YAP1 nuclear transportation, resulting in the downregulation of mi-RNAs and promoting expression of DCLK_1,_ a tumour stem cell marker [237]. CXCR7 regulates CAFs, which are known to drive cancer progression. CXCR7 expression is positively correlated with CAF activation markers in colorectal cancer patients. CXCR7+ CRC cells upregulate miRNAs that cause CAFs to increase their expression of inflammatory cytokines that can trigger EMT [95], allowing the metastasis of CRC cells to the lungs in xenografts. Prokineticin receptors are a family of GPCRs shown to be involved in CRC metastasis. Their activity plays a role in chemotaxis and the production of pro-inflammatory cytokines [238]. Pk-r1 and Pk-r2 are the only receptors in this family and their expression is upregulated in CRC cell lines. Activation of these receptors in CRC cell lines causes a 3–5-fold increase in in vitro metastasis, along with an increase in mRNA and protein levels of metalloproteinases MMP-2,7 and 9. This increase in metastasis was reduced with the addition of an anti-Pk-r2 antibody, suggesting that the Pk-r2 receptor is involved in the metastatic response [239]. In a comparative study, Pk-r2 was shown to be expressed in 45% of human CRC samples and was associated with a high rate of vascularisation and metastasis to the liver and lymph nodes. Pk-r2 expression increased with tumour grade and its expression was negatively correlated with the 5-year survival rate [240]. The use of an antibody against PROK1, the ligand for Pr-k2, is able to reduce the size and amount of liver metastatic lesions in a mouse model for CRC, with immunohistochemistry showing a reduction in the amount of ki-67, a marker of dividing cells [241]. Another of the adhesion GPCR family ADGRG1 (GPR56) is indicated in the progression and metastasis of CRC. ADGRG1 has been shown to be overexpressed in patients with CRC and is associated with a poor prognosis. Overexpression in CRC cell lines promoted migration and invasion via EMT through PI3K/AKT signalling. Knock out of ADGRG1 caused CRC cells to arrest in G0/G1 phase preventing proliferation and reducing EMT markers such as N-cadherin and vimentin [242]. Study of ADGRG1 in patients with CRC, showed that downregulation was indicated with less cell proliferation, migration, and invasion. Those with a higher expression of ADGRG1 had a lower 5-year survival rate, and ADGRG1 expression was found to be a significant prognostic factor for overall survival [243].

### 2.3. Metastatic Lung Cancer

Lung cancer can be divided into small cell and non-small cell lung cancer (NSCLC), the latter making up the vast majority of cases and the prior being more aggressive. Much like most metastatic cancers the chemokine receptor family plays a key role in metastatic lung cancer. CXCR4 is highly upregulated in NSCLCs, and those with the highest expression had a much higher metastatic potential. Overexpression in NSCLC cell lines showed increased migration and invasion, which could be ablated with treatment using anti-CXCR4 antibodies in mice through the prevention of CXCL12 activation of the receptor [244]. The same study also showed that inoculation of lung cancer cells with low CXCR4 expression resulted in far less metastatic clusters than with high-expressing cells. CXCL12-induced migration of NSCLCs was shown to be CXCR4- and not CXCR7-dependent. Knockouts of both were designed in NSCLC lines, and migration was ablated when CXCR4 was knocked out. Xenografts in mice showed that CXCR4 was necessary for metastasis, not CXCR7 [245]. In a meta-analysis study, it was found that CXCR4 was more highly expressed in NSCLC than normal tissue, its expression was higher in later stage cancers as well as in metastatic NSCLC. Patients with higher CXCR4 expression had lower survival rates than those with low expression [246]. One of the treatments for NSCLC is cisplatin therapy, although that can cause long term detrimental effects such as the promotion of pro-metastatic environments. Cisplatin treatment has been shown to reduce tumour size while also increasing secretion of CXCL12, recruitment of metastasis initiating cells and pro-invasive CXCR4+ macrophages, that promote spontaneous metastasis. Cotreatment with a CXCR4 antagonist was able to prevent this metastasis and highlights a route for CXCR4 targeted treatment in NSCLC [247]. The chemokine receptor CXCR2 is described in metastatic lung cancer. In a mouse model of Lewis lung cancer, depletion of CXCR2 resulted in reduced cell proliferation and the rate of spontaneous metastasis [248]. These results were replicated in a model overexpressing CXCR2 with the use of a monoclonal antibody that blocked CXCR2 activation. Equivalent results were shown in the NSCLC lung adenocarcinoma cell line, where knocking out CXCR2 or blocking with a small molecule antagonist decreased invasion and metastasis of cells expressing CXCR2. Samples from humans with lung adenocarcinoma showed that CXCR2 expression was associated with poor prognosis, a history of smoking, as well as RAS pathway activation [249]. In a mouse model of lung cancer overexpressing CXCR2, an increase in the infiltration of TANs was shown, while an inhibition of CXCR2 ameliorated this infiltration as well as increased antitumor T-cell activity, through the promotion of CD+ T cell activation. Much like with CXCR4, cisplatin therapy can lead to CXCR2 mediated immune suppression, and co-treatment with a CXCR2 antagonist was able to show greater antitumor effects than just cisplatin [250].

Lysophosphatidic acid (LPA) receptors are a family of six receptors involved in diverse cellular processes such as cell proliferation, migration, and differentiation [251]. LPA is the endogenous ligand for these receptors, and it is produced when LPC is catalysed by ATX to form LPA. It was shown that the levels of ATX in NSCLC correlated with the tumour stage and grade, suggesting the role of its receptors in lung cancer progression [252]. It was shown that using an LPAR1–4 antagonist was able to reduce cell migration and invasion in vitro, and loss of vasculature and tumour growth in a xenograft model of NSCLC [253]. More recently, the activity of LPA in lung cancer was specifically tied to LPAR1, as this receptor is overexpressed on CAFs, which are known to promote EMT and migration. By silencing LPAR1, CAF proliferation in NSCLC can be reduced, showing the therapeutic potential of targeting LPAR1 for fibrous metastatic lung cancer [171]. LPAR2 contributes to the survival of highly metastatic cell lines to cisplatin treatment via adenylyl cyclase inhibition, whereas LPAR3 was shown to be beneficial in cisplatin treatment [254]. This shows the intricate nature of cell signalling mediated by GPCRs of the same family. GPR78 is an orphan GPCR that is associated with lung cancer metastasis. It is expressed in lung cancer cells and mediates actin stress fibres in a RhoA- and Rac1-dependent manner, thus regulating cell motility. Knocking out GPR78 suppresses cell migration, indicating potential to target GPR78 therapeutically [149]. The use of miRNA-936 was shown to reduce GPR78 expression and was able to regulate NSCLC proliferation, invasion, and migration [255].

### 2.4. Metastatic Prostate Cancer

Prostate cancer is the one of the fastest growing cancers in Europe, and metastatic prostate cancer has a 5-year survival rate of only 30%, with the current treatment generally including hormone therapy, surgical resection, or castration [256]. As with many other cancers, extensive research aims to understand the involvement of GPCRs in order to develop new therapies. CXCR4 signalling is implicated in the development of metastatic prostate cancer. CXCR4 is overexpressed in prostate cancer cells, and its expression correlates with later stage tumours as well as metastasis to both the bones and lymph nodes, a poor prognosis predictor for patients [257]. In prostate cancer, CXCR4 localises to the nucleus where its active signalling could be a mechanism for continuous CXCR4 activation in metastatic prostate cancer [258]. CXCR4 has been shown to interact with PI4KIIIα, a PI4K kinase, and through this interaction on lipid rafts it is able to mediate tumour metastasis, while PI3KIIIα knockouts inhibit CXCR4 mediated prostate cell metastasis [259]. CCR5 signalling is also involved in the metastasis of prostate cancer and is overexpressed in prostate cancer. Activation by its ligand CCL5 induces proliferation and stimulates invasion, which is reduced by a CCR5 antagonist [260]. One of the principal organs for prostate cancer to metastasize to is bone. Studies on CCR5 activation during prostate cancer metastasis in mouse models and treatment with two small molecule inhibitors of CCR5 originally designed for HIV-1 therapy Maraviroc and Vicriviroc, which are CCR5 antagonists, reduced the tumour burden in both the bones and prostate [261]. The tumour suppressor miRNA-455-5p targets CCR5 in prostate cancer, and its overexpression was able to suppress CCR5 mediated proliferation, migration, and induce apoptosis in prostate cancer cells [262].

GPRC6A is an orphan GPCR that has recently gained attention for its role in prostate cancer. *GPRC6A* transcripts are upregulated in prostate cancer, and in prostate cell lines, with ligands to GPRC6A such as calcium and arginine showing a dose-dependent stimulation of ERK activity as well as chemotaxis and proliferation [263]. This dose-dependent response was ablated by silencing GPRC6A. In xenograft models of prostate cancer, cells expressing GPRC6A promoted cell migration and proliferation after stimulation with osteocalcin via ERK and AKT signalling, in comparison to knockout cells [264].

### 2.5. Metastatic Melanoma

Melanoma is another rapidly increasing problem worldwide, and is the fifth most common cancer type in men, and the sixth in women worldwide [265]. Similarly to other metastatic cancers, there are many GPCRs involved in melanoma but of those cancers discussed here, it has the fewest associated receptors identified. As with almost all metastatic cancers, CXCR4 has been definitively identified as a key driver of metastatic melanoma. It was discovered early on that CXCR4 expression in melanoma cells was correlated with poor prognosis and the risk of recurrence was 2.5-fold higher and death 3 times higher than those with low CXCR4 expression [266]. CXCR4 has been shown to assist melanoma metastatis to bones, and exosomes from those cells were able to cause the upregulation of CXCR7 a member of the CXCL12/CXCR4/CXCR7 signalling axis to promote them to a more osteotropic phenotype [267]. In a meta-analysis of melanoma cancer patients, CXCR4 overexpression in melanoma cells was correlated with ulceration, tumour thickness, and lymph node metastasis, and is a strong prognostic biomarker for metastatic melanoma [268]. The use of a CXCR4 antagonist in a murine melanoma model showed antitumor effects that were additive when used in combination with an anti-PDL1 antibody. They also showed a reduction in immunosuppressive regulatory T cells and increasing tumour specific CD8+ cells leading to a reduction in tumour growth [269]. CCR10 is another member of the chemokine family involved in the metastasis of melanoma, with studies in a mouse melanoma model that overexpressed CCR10 showing that these cells had a higher rate of proliferation, the cytoskeleton underwent rearrangement and they had increased migration in response to CCR10 ligands vs. non CCR10 expressing cells [270]. CCR10 expression in melanoma cells was correlated with significantly lower survival time and time to progression, as well as a higher chance of cerebral metastasis [271]. CCR10 influences the immune system, allowing melanoma cells to evade immunosurveillance. T lymphocyte density is inversely correlated with CCR10 expression and lymph node metastasis are shown to have a higher expression of CCR10 [272].

Another family of GPCRs that have been shown to be involved in metastasis are the pyrimidinergic receptors (P2YRs), that participate in the signalling of nucleotides such as ATP and UTP, which have been shown to affect inflammation and the composition of the tumour microenvironment [273,274]. The GPCR P2Y6 has been shown to be involved in other metastatic cancers such as breast cancer [275], and has also been implicated in melanoma. In a mouse model for melanoma transplantation of B16F10 cells in a P2Y6 knockout, there were a significantly reduced number of metastatic lung tumours, and increased survival rates vs. the wild type [276]. Knockout of P2Y6 had no effect on tumour growth, only the ability to metastasize [277]. Expression of P2Y1/2 and 6 in melanoma cells showed that the addition of a P2Y1 agonist reduced cell proliferation and number, while a P2Y2 agonist was shown to increase cell growth and proliferation.

## 3. Biologics Targeting GPCRs in Cancer

Due to the increasingly well-understood role of GPCRs in cancer progression, a large field of work has been developed to identify therapeutic agents to ameliorate the effects of their aberrant expression and signalling, with a shift toward biologics over small molecules in the last decade. To effectively target these receptor structures embedded in the membrane, biologics must be able to specifically target the extracellular region of the receptor, and depending on where they bind in the receptor, they can have different effects on GPCR signalling. In a recent study, Peters et al. proposed an annotation scheme for naming GPCR binding sites clearly and meaningfully. The name of a binding site consists of the GPCR Class, the location (IH: intrahelical EH: extrahelical, IC: intracellular, and EC: extracellular with respect to transmembrane (TM) helices), and the binding site location with respect to the membrane (ext: exterior, mid: middle, and int: interior) (Figure 3). Biologics such as mAbs have a greater specificity and potential efficacy than standard small molecules that allow for precision targeting of GPCRs. There remains a significant unmet need for therapies targeting the vast majority of GPCRs, with very few passing clinical trial stages. In fact, there are only 13 currently approved drugs, of which 10 can be considered biologics (Table 2). Nanobodies, engineered proteins, and peptides are other classes of biologics which are an increasingly promising area for targeting GPCRs, their comparatively smaller size relative to mAbs allow better penetration and access. Engineered proteins and peptides possess a structural design rationale related to the target that can allow for increased efficacy.

### 3.1. Mononclonal Antibodies

Monoclonal antibodies over the last few decades have revolutionised cancer research and therapies. Since the first approved monoclonal antibody, Rituximab in 1997 [294], over 197 antibodies have been approved by the FDA/EMA, and over 90 of those have been indicated for cancer [295]. Trastuzumab (Herceptin) is a mAb that targets the tyrosine kinase receptor HER2. HER2 is overexpressed in 20–30% of breast cancers, and prior to the discovery of trastuzumab, HER2 positive breast cancer had a poor overall survival [296]. This discovery improved the outcome of patients with HER2 positive cancer, although many patients with early-stage breast cancer relapse and those with metastatic breast cancer develop resistance within a decade [296,297]. Another blockbuster mAb is Pembrolizumab (Keytruda) which is a checkpoint inhibitor targeting PD-L1, and is indicated for many cancers such as multiple myeloma and NSCLC [298]. The success of this discovery has led to it being one of the top biologic blockbuster drugs, earning close to 20$ bn USD annually.

To date, there are only three approved mAb treatments targeting GPCRs, of which two are indicated for cancer. Erenumab is a calcitonin gene-related peptide receptor antagonist that is approved for the treatment of migraines [299]. It was found to greatly reduce monthly migraine time and begins its effects within the first week of treatment. It works by preventing binding of the CGRP peptide to the receptor, thereby decreasing vasodilation and inflammation associated with migraines [300]. The other two approved antibodies are Mogamulizumab and Talquetamab. Mogamulizumab is an anti-CCR4 mAb that has been approved for the treatment of T-cell lymphomas mycosis fungoides and Sézary syndrome, two of the most common T-cell lymphomas [301]. Prior to the discovery of Mogamulizumab, the only treatment was allo-HSCT, which has a high morbidity with overall survival being between 30 and 40% [281]. In a phase 3 international trial, Mogamulizumab was compared to Vorinostat, a standard treatment for T-cell lymphoma, in patients with early-stage mycosis fungoides. It was found that Mogamulizumab had a median progression-free survival of 6.7 months compared to 3.8 months in the Vorinostat group and had a higher proportion of patients who had an overall response [302]. Despite these promising results, patients can eventually develop resistance to Mogamulizumab treatment. Resistance usually develops as patients lose the target antigen CCR4, rendering Mogamulizumab ineffective, but there is another unknown mechanism of resistance in which patients retain high expression of CCR4 [303]. Talquetamab is a bispecific mAb that targets CD3 and GPRC5D and was approved for the treatment of multiple myeloma in August 2023 [304]. Most patients with multiple myeloma relapse and those who relapse have poor overall survival of roughly 12% [280]. Talquetamab can bind to GPRC5D, a biomarker associated with high-risk myeloma, and CD3, and induces T-cell mediated death of myeloma cells expressing GPRC5D via recruitment and maturation of T-cells [305]. In a phase I/II study of patients with triple and penta-refractory multiple myeloma, Talquetamab showed an overall response rate of around 70% up to 18 months after treatment. Interestingly, results were similar for the cohort who had previously received other bispecific antibody or CAR T treatments, suggesting the potential use of this in combination with those treatments to overcome resistance [306,307]. One caveat is that almost all patients had adverse effects of grade 3 or higher, although none died, with the most common being cytokine-release syndrome and infections [307]. Another mAb of interest that has not yet received approval but has reached late-stage clinical trials for its potential use in HIV and COVID-19 treatment is Leronlimab. Leronlimab is a CCR5 antagonist mAb. Leronlimab is currently in phase III clinical trials for preventing HIV infection [308], but has previously shown promise in treating breast cancer. In triple negative breast cancer lines, Leronlimab was shown to reduce migration, calcium signalling, as well as enhance the effect of doxorubicin in killing breast cancer cells. Furthermore, in xenograft mice models it was able to reduce tumour burden of > 95% after 6 weeks of treatments [309].

Leronlimab has also shown success in early clinical trials; a phase I trial showed that it was well-tolerated in combination with carboplatin and showed early signs of anti-tumour activity [310]. In a basket study of advanced and metastatic solid tumours, Leronlimuab showed a median progression free survival of 6 months in greater than 75% of patients, along with a reduction in circulating tumour associated cells [311].

Monoclonal antibodies have revolutionised the oncology field and have become the gold standard of care in many cases, yet advancements have yet to be fully realised with GPCR targets, with many of the promising therapies failing due to adverse off target effects, lack of efficacy, or development of resistance.

### 3.2. Protein/Peptides

Protein and peptide therapies are the largest group of biological molecules targeting GPCRs. From hormone replacement to engineered protein analogues and mimetics, small polypeptides have become a staple in treating many diseases. For proteins and peptides, the most well-known success story for targeting GPCRs is in the treatment of type 2 diabetes via GLP-1R agonists. The first of these being exenatide, a synthetic peptide that naturally occurs in lizards’ salivary glands. It has a 53% amino acid sequence identity with GLP-1, the natural ligand for GLP-1R, but has a greater than 1000-fold potency for the receptor. In phase 3 clinical trials, roughly 40% of patients had a reduction in HbA1c levels of ≤7% [312]. This was then approved for the treatment of type 2 diabetes in 2005, although it has been gradually phased out due to the emergence of superior protein therapies such as liraglutide, and semaglutide. These therapies are GLP-1 analogues that have superior half-life and potency for the receptor both showing greater glucose lowering and weight loss effects, leading to their approval in 2009, and 2017, respectively [313].

In terms of targeting GPCRs for use in cancer therapies there are eight therapies currently approved and many in clinical trials. Those that are approved mainly target three different receptor groups, gonadotropin-releasing hormone receptors, somatostatin receptors, and glycoprotein hormone receptors. Abaraelix is the first synthetic decapeptide GnRH antagonist developed, and is approved for the use in advanced prostate cancer. It works by inhibiting the activation of GnRH, preventing the secretion of LH and FSH, which thereby reduces testosterone levels, a key driver of prostate cancer. It was first approved in 2004 due to its ability to achieve medical castration quickly and well-tolerated without having testosterone flare ups which can impede treatment [314]. Lanreotide is a somatostatin analogue and is indicated for use in locally or metastatically advanced neuroendocrine tumours and is the only currently approved protein/peptide therapy for metastatic cancer [315]. The last class of currently approved protein/peptide therapies is goserelin, which in men is used for prostate cancer, and in women is used for breast cancer treatment. It is a synthetic analogue of luteinizing hormone-releasing hormone and antagonises LsHR to prevent the secretion of both testosterone and oestrogen [316]. The only peptide currently approved that does not target one of the three previously mentioned receptors is Motixafortide, which recently gained approval in September 2023 for its use in autologous stem cell transplantation in patients with multiple myeloma. Motixafortide is a cyclic synthetic peptide CXCR4 antagonist that causes haematopoietic stem and progenitor cells to mobilise rapidly and for a sustained duration. It was shown in a phase 3 trial in combination with G-CSF to increase the amount of mobilising CD34+ cells vs. G-CSF alone after just one apheresis, 92.5% vs. 26.2%, respectively [317]. It has also shown some promise for the treatment of metastatic diseases. In a phase II trial for metastatic pancreatic ductal adenocarcinoma (mPDAC) it was shown in use with pembrolizumab and chemotherapy to be well-tolerated and showed signs of efficacy in an aggressive disease [318]. It is now being assessed in another phase II trial for metastatic pancreatic dual adenocarcinoma where they are testing its effect on progression free survival [319]. Although there are few approved protein/peptide therapies, there are many more promising ones in clinical trials. One such example is Ctce-9908 (PTX-9908) a CXCR4 antagonist an analogue of its ligand CXCL12. It was shown to reduce tumour burden in a mouse model of breast cancer seven-fold, as well as greatly reducing metastasis [320]. It was then shown in a phase I/II trial for solid tumours to be well-tolerated, and showed early signs of efficacy in ovarian cancer, and then in 2005, it was granted orphan drug status for young adults with osteosarcoma [321,322]. Currently recruiting for a phase I/II trial for patients with non-resectable hepatocellular carcinoma [323].

### 3.3. Nanobodies

Nanobodies are a unique class of biologics, derived from camelid antibodies. Caemilds produce a class of unique antibody consisting of only heavy chains, and it is this single variable antigen-binding (VHH) fragment that makes up a nanobody [324]. Nanobodies have distinct structural characteristics that give them an advantage over monoclonal antibodies. Their small size (~15 kDa), convex shape, and their extended CDR3 allow them to bind onto what would be classically considered obstructed structures that mAbs would be unable to reach, giving them exclusive access to targeting these sites [324,325]. They are also quite stable and resistant to harsh conditions such as pH and heat; this gives them potential use in the tumour microenvironment as well as in combination with radiotherapy [326].

Due to these useful characteristics, research into their use as therapeutics has grown significantly over the last 20 years. This has resulted in the approval of the first nanobody-based therapy in Caplacizumab for the treatment of acquired thrombotic thrombocytopenic purpura (TTP). It works by targeting the von-Willebrand factor and preventing its interaction with platelet glycoprotein receptors. It was shown in a phase 3 trial to reduce the time for platelet normalisation, the incidence of TTP-related death as well as recurrence during the trial [327]. Recently another nanobody-based therapy was approved for the treatment of cancer. Ciltacabtagene, which is a chimeric antigen receptor (CAR) T-cell therapy that employs nanobodies as the targeting domain rather than the usual scFv domain [328]. It has been approved for use in patients with relapsed/refractory multiple myeloma after showing in clinical trials an overall response rate of 97.8% with a duration of response of 21.8 months [329]. As of yet, there are no approved nanobody-based therapies that target GPCRs, in fact there are currently only three nanobody-based therapies that have undergone clinical trials for targeting GPCRs and only one of them was for use in cancer therapy. The first to enter clinical trials was ALX-0651, a biparatopic anti-CXCR4 nanobody for use in cancer therapy that was selected from a library generated from peripheral blood mononuclear cells of llamas that were immunised with HEK293T cells expressing CXCR4. From this library, two nanobodies were selected and joined via GGGGS linker to form ALX-0651. In HIV models, it impeded CXCR4-mediated entry of HIV into MT-4 cells, and in monkeys it was able to mobilise stem cells in a comparable manner to plerixafor, an approved CXCR4 antagonist [330,331]. It was terminated after phase I clinical trials, as although it was well-tolerated and effective, preclinical data suggested that it would not surpass current standard care [332,333]. The second nanobody targeting GPCRs in clinical trials is the Anti-CXCR2 Biparatopic nanobody, currently being developed by Novartis for use in inflammatory disorders [334]. Biparatopic antibodies have already shown some preclinical promise as they were able to produce monovalent antibodies targeting CXCR2 that could selectively target and inhibit the activation of CXCR2 through both CXCL1 and CXCL8 binding. A biparatopic version was created by combining the top two candidates that bound distinct epitopes and showed that this increased the overall potency of the nanobody [335]. Finally, BI 665088 is a bivalent nanobody that targets CX3CR1. It was developed from a library of PBMCs from llamas immunised with CX3CR1 DNA, then followed by immunisation with Caki cells overexpressing CX3CR1, then immunisation with peptides derived from the extracellular loops of CX3CR1. From this library the top four lead candidates were chosen from competitive binding assays and turned into bivalent constructs from which BI 665088 emerged as the most promising. In murine atherosclerosis models, BI 665088 was able to reduce aortic plaque formation by 62% in 6 weeks, showing for the first time the effect of a CX3CR1 antagonist in vivo [336]. It has since been shown in phase I clinical trials to be well-tolerated in humans, with little to no adverse effects [337].

Despite the potential of nanobodies, they are yet to show any impact in the therapeutic targeting of GPCRs, although where they have been able to make an impact is in their use in stabilising GPCRs for X-ray crystallography and Cryo-EM, allowing for structural determination of various receptors and the mapping of their binding regions. This has been performed by developing nanobodies that recognise the intracellular parts of GPCRs, allowing them to stabilise GPCR conformations allowing for the generation of agonist-bound GPCR crystal structures [338]. This gives us valuable insight into the activation mechanism of GPCRs and can allow for structure lead drug design.

## 4. Conclusions

The targeting of G protein-coupled receptors (GPCRs) in metastatic cancer presents a promising frontier in cancer therapeutics. As integral players in cell signalling, GPCRs engage in many key processes that promote tumour growth, invasion, and metastasis, such as angiogenesis, immune modulation, and cell migration. Notably, GPCRs from the chemokine receptor family such as CXCR4, CXCR2, CCR7 are of great interest due to their role in tumorigenesis, but many other receptors as shown have a role such as receptors from the LPA and frizzled receptor families. Despite noteworthy progress, there are still several challenges to overcome in the development of biologics targeting GPCRs. The complex structure of GPCRs, which often includes multiple ligand-binding sites and the potential for biased signalling, as well as the formation of oligomers and receptors homo/heterodimerizing, complicates drug design. Additionally, the widespread expression of GPCRs across various tissues poses a risk for off-target effects, raising safety concerns. Therefore, a more refined understanding of GPCR signalling dynamics and tissue-specific receptor expression is essential for improving therapeutic precision and minimising adverse effects, and translating promising pre-clinical data into working therapeutics. The use of nanobodies and their ability to stabilise GPCRs for structural and functional analysis is one such method to help elucidate these issues. Nanoparticles/carrier systems also show promise in this area. These are colloidal nano-scale systems capable of carrying small molecules as well as larger macromolecules such as genes or proteins. These can protect biologics from the in vivo environment, preventing early degradation and accumulation in non-specific areas. They also can increase accumulation in tumours leading to greater cytotoxic effects [339]. Examples of these include liposomes similar to a cell membrane with a hydrophilic core and hydrophobic shell facilitating passive targeting, biomimetics which include cell membranes, extra cellular vesicles and viruses allowing evidence of the immune system and long circulation times, and lastly polymeric nanoparticles, which use alginate or gelatine to make nanogel spheres, yet these are still in early development with pharmacokinetics and biosafety still unclear [340].

The successes achieved with GPCR-targeted biologics, such as the inhibition of CXCR4/CCR4 by monoclonal antibodies, demonstrate the therapeutic potential of these receptors. However, the heterogeneity of GPCR expression in different tumour microenvironments necessitates further exploration of context-dependent targeting strategies. For future research, more in-depth studies into GPCR signalling bias are needed. Some GPCRs can signal through multiple intracellular pathways, with certain pathways being more oncogenic than others. By developing biologics that selectively target harmful signalling routes (biased agonism or antagonism), it may be possible to minimise off-target effects, while maximising therapeutic efficacy.

In conclusion, while the development of biologics targeting GPCRs in metastatic cancer has shown promise, there is still a need for more targeted, selective approaches to fully exploit their therapeutic potential. Continued research into GPCR signalling mechanisms and the TME will be crucial for translating these biologics into effective clinical treatments. As our understanding of GPCR biology expands, so too will the opportunities for novel interventions, potentially transforming the landscape of metastatic cancer treatment.

## Figures and Tables

**Figure 1 biotech-14-00007-f001:**
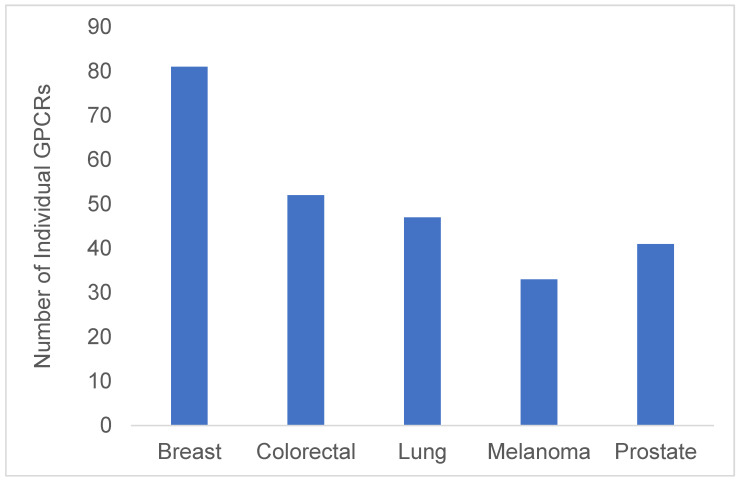
Number of GPCRs involved in the top five most common metastatic cancers.

**Figure 2 biotech-14-00007-f002:**
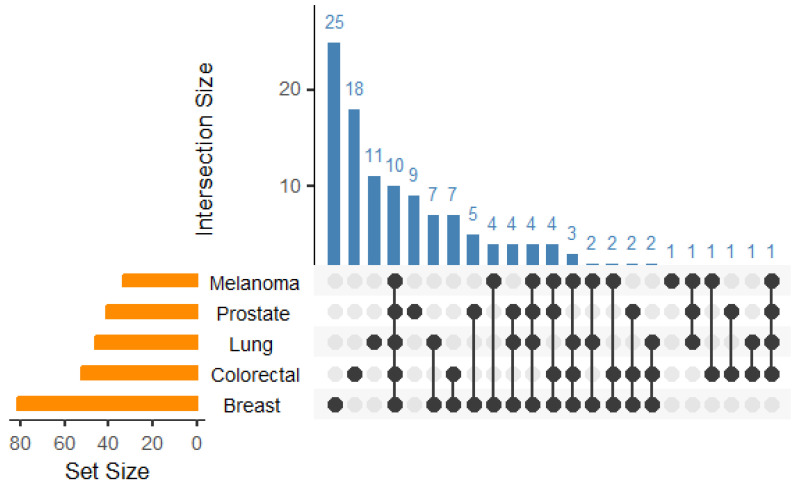
Upset plot showing the overlap of different GPCRs in the top five most common metastatic cancers. Histogram in blue (top) showing the distribution of individual GPCRs across all five cancer types. Bar chart in amber (bottom left) showing total numbers of receptors per cancer type. Black spheres linking both datasets. Data taken from Table 1.

**Figure 3 biotech-14-00007-f003:**
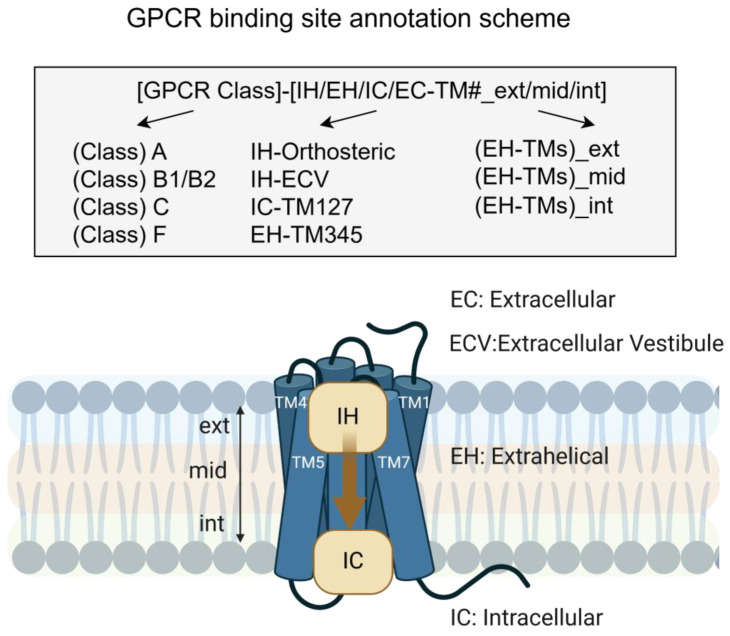
Schematic representation of GPCR structure in the cell membrane with ligand binding sites annotated. (A) GPCR binding site annotation starts with GPCR Class (A, B1, C) followed by position with respect to the transmembrane domain (EC: extracellular, IH: intrahelical, ECV: extracellular vestibule, IC: intracellular, and EH: extrahelical). For extrahelical binding sites, the annotation is tagged with membrane position along the vertical axis (ext: exterior, mid: middle, or int: interior). Adapted from Peters et al., 2024 [278].

**Table 1 biotech-14-00007-t001:** Table of the GPCRs involved in the most common metastatic cancers. Column one represents the name of the receptor, and column two represents the type of metastatic it is involved in.

Receptor	Metastatic Cancer	Reference
5-HT1A	Prostate	[22]
5-HT1D	Colorectal	[23]
5-HT2B	Colorectal	[24]
5-HT4	Prostate	[25]
5-HT7	Breast, Lung	[26,27]
A2BR	Breast, Colorectal, Lung, Melanoma	[28,29]
A3R	Breast, Colorectal	[30]
ACKR2	Breast, Lung	[31,32]
ADGRE1	Colorectal	[33]
ADGRF5	Breast, Colorectal	[34,35]
ADGRG1	Breast	[36]
ADRA2A	Breast	[37]
ADRA2C	Breast	[37]
ADRB2	Breast, Colorectal	[37,38]
ADRB3	Lung	[39]
APNLR	Breast, Lung, Prostate, Melanoma	[40,41,42]
AVPR1A	Prostate	[43]
C3AR1	Breast, Melanoma	[44,45]
C5AR1	Breast, Colorectal, Melanoma,	[45,46,47]
CASR	Breast, Prostate	[48]
CB2	Breast, Lung, Prostate	[49,50]
CCKAR	Lung	[51]
CCR1	Breast, Colorectal, Lung, Melanoma, Prostate	[52,53,54]
CCR2	Breast, Colorectal, Lung, Melanoma, Prostate	[55,56]
CCR3	Breast, Colorectal, Melanoma, Prostate	[57,58,59,60]
CCR4	Breast, Colorectal, Melanoma, Prostate	[61,62,63]
CCR5	Breast, Colorectal, Lung, Melanoma, Prostate	[64,65]
CCR6	Breast, Colorectal, Lung, Melanoma, Prostate	[66,67,68,69]
CCR7	Breast, Colorectal, Lung, Melanoma, Prostate	[66,70]
CCR8	Breast, Colorectal, Lung, Melanoma	[66,71]
CCR9	Breast, Lung, Melanoma, Prostate	[66,72]
CCR10	Breast, Lung, Melanoma	[73,74]
CCRL2	Colorectal, Prostate	[75,76]
CRHR1	Prostate	[77]
CX3CR1	Breast, Lung, Prostate	[78,79,80]
CXCR1	Breast, Colorectal, Lung, Melanoma, Prostate	[81,82,83]
CXCR2	Breast, Colorectal, Lung, Melanoma, Prostate	[81,83,84]
CXCR3	Breast, Colorectal, Lung, Melanoma, Prostate	[85,86,87]
CXCR4	Breast, Colorectal, Lung, Melanoma, Prostate	[88,89]
CXCR5	Breast, Lung, Melanoma, Prostate	[90]
CXCR6	Breast, Lung, Melanoma, Prostate	[91,92,93]
CXCR7	Colorectal, Lung, Melanoma, Prostate	[94,95,96,97]
EDNRA	Colorectal	[98]
EDNRB	Breast, Melanoma	[99,100]
F2R	Breast, Colorectal Melanoma, Prostate	[101,102,103]
FFAR1	Breast, Prostate	[104,105]
FPR1	Breast, Colorectal, Lung, Melanoma	[106,107,108,109]
FPR2	Breast, Colorectal	[106,110]
FSH	Breast, Lung, Prostate	[111,112]
FZD1	Breast	[113]
FZD2	Breast, Colorectal, Lung	[114]
FZD5	Prostate	[115]
FZD7	Breast, Colorectal, Melanoma,	[116,117,118]
FZD8	Breast, Colorectal, Prostate	[119,120,121]
GABBR2	Breast	[106]
GALR1	Colorectal	[122]
GNRHR	Breast, Colorectal, Prostate	[123,124]
GPER	Breast	[125]
GPR107	Prostate	[126]
GPR132	Breast	[127]
GPR141	Breast	[128]
GPR15	Colorectal	[129]
GPR161	Breast	[130]
GPR171	Breast, Lung	[131,132]
GPR176	Colorectal	[133]
GPR18	Melanoma	[134]
GPR19	Breast, Melanoma	[135,136]
GPR31	Colorectal	[137]
GPR34	Colorectal	[138]
GPR35	Colorectal	[139]
GPR37	Lung	[140]
GPR39	Breast, Prostate	[141,142]
GPR4	Colorectal, Melanoma	[143,144]
GPR50	Breast	[145]
GPR55	Breast	[146]
GPR65	Colorectal	[147]
GPR75	Prostate	[148]
GPR78	Lung	[149]
GPRC6A	Breast, Prostate	[150,151]
GRM3	Breast	[152]
GRPR	Colorectal	[153]
GSHR	Lung, Melanoma, Prostate	[154,155,156]
HCAR1	Breast	[157]
HRH1	Breast,	[158]
HRH3	Breast, Lung	[159,160]
HTR2B	Colorectal	[161]
LGR4	Breast, Lung, Prostate	[162,163,164]
LGR6	Breast, Colorectal, Lung	[165,166,167]
LH	Breast, Colorectal	[168,169]
LPAR1	Breast, Lung, Melanoma	[170,171,172]
LPAR2	Breast	[173]
LPAR3	Breast	[174]
LPAR5	Breast	[175]
LPAR6	Breast	[176]
LTB4R	Breast	[177]
M2R	Colorectal, Lung	[178,179]
M3R	Lung	[180]
MRGD	Lung	[181]
NMUR1	Colorectal	[182]
NMUR2	Colorectal	[183]
NPY1R	Breast, Colorectal, Melanoma, Prostate	[184]
NPY5R	Breast	[185]
NTSR1	Breast, Lung	[186,187]
OPKR1	Breast	[188]
OPN3	Lung	[189]
OXER1	Breast, Prostate	[150,190]
OXTR	Breast, Melanoma	[191,192]
P2YR1	Lung	[193]
P2YR11	Breast	[194]
P2YR6	Lung	[195]
PROK1	Colorectal	[196]
PROK2	Colorectal	[197]
PTAFR	Breast	[198]
PTGER1	Colorectal	[199]
PTGER2	Prostate	[200]
PTH1R	Breast, Lung	[201,202]
QRFPR	Prostate	[203]
RXFP1	Breast	[204]
S1PR1	Breast, Colorectal	[205,206]
S1PR3	Breast, Colorectal	[206,207]
SUNCR1	Lung	[208]
TACR1	Breast	[209]
TACR2	Lung	[210]
TBXA2R	Breast, Colorectal, Lung, Melanoma, Prostate	[211]
XCR1	Breast, Lung	[212,213]

**Table 2 biotech-14-00007-t002:** Table of approved drugs targeting GPCRs for cancer treatment. Sorted by drug type including monoclonal antibodies, peptides, and small molecule drugs, showing their target, mechanism, and indication.

Name	Target Receptor	Drug Type	Status	Mechanism	Type of Cancer	Reference
Mogamulizumab	CCR4	Monoclonal Antibody	Approved	Antagonist	Mycosis fungoides/Sezary syndrome	[279]
Talquetamab	GPRC5D	Bispecific Antibody	Approved	Agonist	Multiple Myeloma	[280]
Motixafortide	CXCR4	Peptide	Approved	Antagonist	Hematopoietic Stem Cell Mobilisation in Multiple Myeloma	[281]
Goserlin	GNRHR	Synthetic Peptide	Approved	Agonist	Advanced Prostate/Breast Cancer	[282]
Lanreotide	SSR2	Synthetic Peptide	Approved	Agonist	Metastatic/Advanced Pancreatic Neuroendocrine Tumours	[283]
Abralelix	GNRHR	Synthetic Peptide	Approved	Antagonist	Advanced Prostate Cancer	[284]
Leuprolide	GNRHR	Synthetic Peptide	Approved	Agonist	Advanced Prostate Cancer	[285]
Degarelix	GNRHR	Synthetic Peptide	Approved	Antagonist	Advanced Prostate Cancer	[286]
Histrelin	GNRHR	Synthetic Peptide	Approved	Agonist	Advanced Prostate Cancer	[287]
Triptorelin	GNRHR	Synthetic Peptide	Approved	Agonist	Advanced Prostate Cancer	[288]
Vismodegib	SMO	Small Molecule	Approved	Antagonist	Metastatic/Advanced Basal Cell Carcinoma	[289,290]
Sonidegib	SMO	Small Molecule	Approved	Antagonist	Locally Advanced Basal Cell Carcinoma	[291]
Plerixafor	CXCR4	Small Molecule	Approved	Antagonist	Hematopoietic Stem Cell Mobilisation in Multiple Myeloma/Non-Hodgkins Lymphoma	[292,293]

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
