# Peer review of "The Use of Biologics for Targeting GPCRs in Metastatic Cancers"

_biotech, 2025, doi:10.3390/biotech14010007_

Round 1
Reviewer 1 Report
Comments and Suggestions for Authors
This article examines the role of G protein-coupled receptors 6 (GPCR) in cancer metastasis and investigates the potential for developing biotherapeutic agents against these receptors. The focus is primarily on the involvement of GPCRs in five prevalent metastatic cancers: lung, breast, colorectal, melanoma, and prostate cancer. Among the approximately 390 therapeutically relevant GPCRs, 125 have been identified as promoting metastatic disease within these cancer types. GPCRs play a crucial role in numerous facets of the metastatic phenotype, including known signaling of chemokine receptors and emerging data on orphan receptors. Despite extensive information on receptor targets, only thirteen therapies for metastatic cancers have been approved—three small molecules and the rest consisting of synthetic and non-synthetic peptides or monoclonal antibodies. This article will discuss both existing and potential applications of monoclonal antibodies, proteins, peptides, and nanobodies as therapeutics targeting GPCRs in metastatic cancer.
I recommend that this manuscript be accepted pending minor revisions, as outlined below.
1. The authors are encouraged to provide a concise overview of the impact of GPCR on the metastasis of various types of tumors in the introduction section.
2. The author should address the limitations inherent in this field within the conclusion section and delineate the potential avenues for future research and development.
3. Where is Chapter two?
4. The author should provide a comprehensive overview of fundamental tumor characteristics, including mechanisms of cancer metastasis, in the introduction section.
5. The authors should provide a concise overview of the impacts of epithelial-mesenchymal transition and fibroblasts on tumor metastasis.
6. It is recommended to cite the following literature:
[1] A. Gu, J. Li, M.-Y. Li, Y. Liu, Patient-derived xenograft model in cancer: establishment and applications. MedComm, 2025, DOI: 10.1002/mco2.70059
[2] T. Zhou, H. Yan, Y. Deng, Y. Zhu, X. Xia, W. Wu, W.-H. Zhang, H.-N. Chen, J.-K. Hu, Z.-G. Zhou, Y. Shu, Y. Li, H. Xu, The role of long non-coding RNA Maternally Expressed Gene 3 in cancer-associated fibroblasts at single cell pan-cancer level. Interdiscip. Med. 2024, 2, e20240018. https://doi.org/10.1002/INMD.20240018
[3] Wang J, Liao Z-X. Research progress of microrobots in tumor drug delivery. Food & Medicine Homology, 2024, 1(2): 9420025. https://doi.org/10.26599/FMH.2024.9420025
Author Response
We thank the reviewer for the very helpful suggestions on improving the manuscript.
Reviewer 1 comment “The author should provide a comprehensive overview of fundamental tumor characteristics, including mechanisms of cancer metastasis, in the introduction section.”
We have now indicated some of the GPCR mediated signalling pathways commonly involved in cancer and a section on EMT and CAFs.
Reviewer 1 comment “The authors should provide a concise overview of the impacts of epithelial-mesenchymal transition and fibroblasts on tumor metastasis.”
Included a section briefly underlying the role in tumour progression
Reviewer 1 “ The author should address the limitations inherent in this field within the conclusion section and delineate the potential avenues for future research and development.”
This is now addressed from line 575 in the conclusions
Reviewer 1: “It is recommended to cite the following literature:"
We have added the recommended reference Wang J, Liao Z-X. Research progress of microrobots in tumor drug delivery. Food & Medicine Homology, 2024, 1(2): 9420025. https://doi.org/10.26599/FMH.2024.9420025
Reviewer 2 Report
Comments and Suggestions for Authors
Dear Authors;
Re: [Manuscript ID: biotech-3407994]
Title: "The Use of Biologics for Targeting GPCRs in Metastatic Cancers"
In this manuscript, which is in the form of a review article, you aimed to cover studies describing the role of G-protein coupled receptor (GPCR) behaviour contributing to metastasis in cancer and the developments of biotherapeutic drugs towards precise tumor targeting.
It is a very important literature bringing to the attention of readership recent advancements in cancer therapy.
Please consider the following suggestions / comments:
A List of Abbreviations will benefit the readers significantly.
A 31-page comprehensive review deserves more than 2 illustrations and 2 Tabulations.
An image depicting the general structure / components of a typical GPCRs will be useful for readers.
Please describe the definition of "nanobody" in the introduction briefly.
The sentence in the Abstract (Lines 16-18), "In contrast to the abundant information ..." needs re-phrasing.
In the Introduction, please provide more information on "small molecules to large proteins".
Use of nanoparticles / nanocarrier systems, such as Nanoliposomes or Tocosome, in modern cancer therapy and precise targeting can be depicted in an image (or a Table).
Please describe the abbreviations used in the Figures and Tables.
The Copyright situations of Figures need to be clarified.
Data presented in the Figures seem to need error bars.
Datasets mentioned in Figure 2 are not covered / described in the text.
Two sentences in Line 458 are joined together: "... sites [323,324]They’re ...". Please check and amend.
Author Response
We thank the reviewer for the very helpful recommendations to improve the manuscript.
Reviewer 2 comment “The sentence in the Abstract (Lines 16-18), "In contrast to the abundant information ..." needs re-phrasing.”
Now rephrased & highlighted
Reviewer 2: In the Introduction, please provide more information on "small molecules to large proteins". -
We have now provided better detail here
Reviewer 2 comment “Please describe the definition of "nanobody" in the introduction briefly.
This is now added
Reviewer 2 “Data presented in the Figures seem to need error bars.”
Error bars are not required for this graph
Reviewer 2: “Datasets mentioned in Figure 2 are not covered / described in the text.” This is clarified here
Reviewer 2: “A 31-page comprehensive review deserves more than 2 illustrations and 2 Tabulations.
An image depicting the general structure / components of a typical GPCRs will be useful for readers.”
We have now added the suggested figure, Figure 3, adapted with permission of the authors of a recent study
Reviewer comment 2 “Two sentences in Line 458 are joined together: "... sites [323,324]They’re ...". Please check and amend. ”
Now fixed and ref numbers updated to [326,327] line 516 due to earlier updates in text
Reviewer 2 comment “Use of nanoparticles / nanocarrier systems, such as Nanoliposomes or Tocosome, in modern cancer therapy and precise targeting can be depicted in an image (or a Table). ”
Described in the text in greater detail from line 585, as a future prospect for improving biologic drug delivery